# Mild Oxidation of Organosulfur Compounds with H2O2 over Metal-Containing Microporous and Mesoporous Catalysts

**Vasile Hulea [1,*], Emil Dumitriu [2] and François Fajula [1]**

[1] Institut Charles Gerhardt Montpellier, UMR 5253 CNRS, Université de Montpellier, ENSCM, 240, Avenue du Professeur Emile Jeanbrau, CEDEX 05, 34296 Montpellier, France; francois.fajula@enscm.fr

[2] Faculty of Chemical Engineering and Environmental Protection, "Gheorghe Asachi" Technical University of Iasi, 71 D. Mangeron Ave., 700050 Iasi, Romania; edumitri@ch.tuiasi.ro

[*] Correspondence: vasile.hulea@enscm.fr

**Abstract:** Mild catalytic oxidation of thioethers and thiophenes is an important reaction for the synthesis of molecules with pharmaceutical interest, as well as for the development of efficient processes able to remove sulfur-containing pollutants from fuels and wastewater. With respect to the green chemistry principles, hydrogen peroxide ($H_2O_2$) is the ideal oxidant and the Me-containing porous materials (Me = Ti, V, Mo, W, Zr) are among the best heterogeneous catalysts for these applications. The main classes of catalysts, including Me-microporous and mesoporous silicates, Me-layered double hydroxides, Me-metal–organic frameworks, are described in this review. The catalytic active species generated in the presence of $H_2O_2$, as well as the probable oxidation mechanisms, are also addressed. The reactivity of molecules in the sulfoxidation process and the role played by the solvents are explored.

**Keywords:** liquid-phase oxidation; hydrogen peroxide; metal-silicates; metal-layered double hydroxides; metal–organic frameworks

## 1. Introduction

Liquid phase oxidation of organic molecules is widely used in industrial processes. Usually, these processes use organic peracids and inorganic compounds in stoichiometric amounts as oxidizing agents. As a result, numerous by-products and large volumes of liquid wastes are generated. To overcome these drawbacks, catalytic methods based on environmentally friendly oxidizing agents, such as hydrogen peroxide ($H_2O_2$), peroxides or hydroperoxydes, have been considered. Significant research efforts have been paid to develop cleaner processes based on the use of homogeneous or heterogeneous oxidation catalysts. Hydrogen peroxide is an ideal oxidant owing to its high effective-oxygen content (47%), cleanliness (water being the only by-product), good oxidation potential ($E_o$ = 1.763 V at pH 0, and $E_o$ = 0.878 V at pH 14) and acceptable safety in storage and operation [1]. To enhance its reactivity under mild conditions, $H_2O_2$ needs homogeneous or heterogeneous catalysts. With respect to the green chemistry principles, the heterogeneous catalysis better meets the requirements of sustainable chemistry [2–4]. The discovery in 1983 of titanium silicalite (TS-1, an MFI-type molecular sieve) has opened up a new era for cleaner oxidation technologies using $H_2O_2$ as oxidant and a metal-based solid catalyst [5,6]. Many materials, including framework-substituted microporous and mesoporous molecular sieves, layered double hydroxide-type materials, encapsulated metal complexes, peroxometalates and metal-organic frameworks have been reported in recent decades as suitable catalysts for liquid phase oxidation with $H_2O_2$ [7–14]. Among them, catalysts containing Ti, W, Mo and V demonstrate high activities for the mild oxidation of various substrates including olefins, phenols, paraffins, aromatics, amines, alcohols, thioethers and sulfoxides [15–20].

Catalytic oxidation of thioethers (organic sulfides) with $H_2O_2$ is an important reaction, notably for synthesizing molecules of pharmaceutical interest [21–24]. On the other hand, many studies on the oxidation of thiophenes and dimethyl sulfoxide (DMSO) have been performed over the last decades. The aim was to develop efficient processes for removing sulfur-containing pollutants from fuels and wastewaters (Table 1).

**Table 1.** Model sulfoxidation reactions and their applications.

| Reaction | Potential Application |
| --- | --- |
|  | Sulfoxides and sulfones synthesis |
|  | Fuel catalytic oxidative desulfurization |
|  | DMSO removal from wastewater |

Organic sulfur compounds are particularly undesirable in liquid hydrocarbon fuels. Their presence has been associated with the corrosion of refining equipment and with the premature breakdown of combustion engines. Sulfur also poisons many catalysts that are used in the refining or in the conversion of gaseous effluents. Moreover, the formation of $SO_x$ during combustion is responsible for environmental problems (e.g., acid rain, ozone and smog generation) and respiratory disorders.

Catalytic hydrodesulfurization (HDS) is industrially used to reduce the sulfur content of the liquid products. HDS is an expensive process which requires high hydrogen pressure. On the other hand, molecules such as benzothiophene, dibenzothiophene and their derivatives constitute very refractory molecules in this process. In order to eliminate undesirable sulfur compounds or to convert them into more innocuous forms, processes different from HDS have been employed. Among them, the catalytic oxidative desulfurization (called ODS) is a promising alternative technology to the current hydrodesulfurization process [25–32]. In ODS, the sulfides and thiophenes are oxidized with $H_2O_2$ or ROOH into their corresponding sulfoxides and sulfones, which are preferentially extracted due to their increased polarity [26].

Selective catalytic sulfoxidation was also proposed as an effective method for the processing of wastewaters contaminated by dimethylsulfoxide, by converting it into biodegradable dimethylsulfone [33,34]. All these reactions are carried out under mild conditions in the liquid phase and $H_2O_2$ is the most widely used oxidizing agent.

Some reviews have been devoted to the sulfoxidation reaction. Most of them relate to ODS processes, dealing in particular with the mass transfer between the oil and the polar phase and the reaction-extraction combination [28–31,35–37].

The aim of this review is to describe some relevant scientific aspects concerning the catalysts and the oxidation of thioethers/thiophenes with $H_2O_2$. The following points are covered below: (i) the reactivity of molecules in sulfoxidation; (ii) the catalysts (Me-microporous and mesoporous materials, Me-layered double hydroxides (LDH), Me-metal organic frameworks (MOF); (iii) the active species and mechanism in the oxidation with $H_2O_2$; (iv) the role of the solvent. Our major contributions in the field are underscored in this review.

## 2. Reactivity of Molecules in Sulfoxidation

It is well known that the oxidation of organic sulfides by hydrogen peroxide occurs via a heterolytic process involving the nucleophilic attack of the sulfur atom on the oxygen atom (Scheme 1) [38,39].

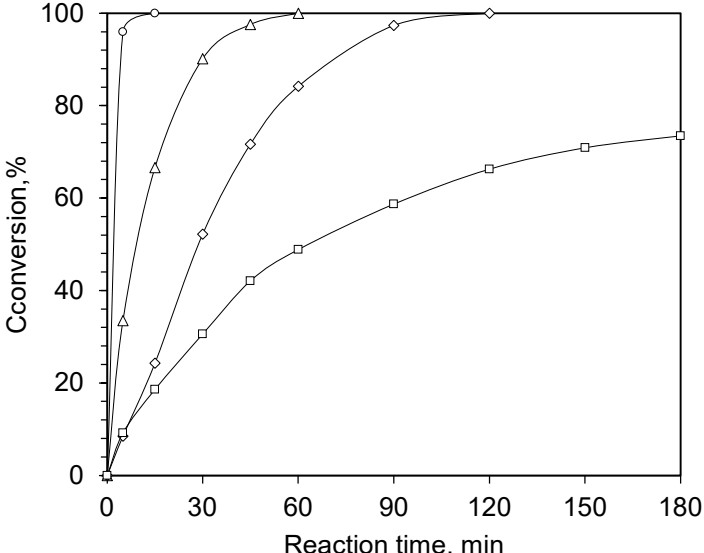

**Scheme 1.** Nucleophilic attack of the sulfur atom on the oxygen atom.

Accordingly, the reactivity of different molecules in this reaction depends on the electronic density on the sulfur atom. For example, the electron densities of the sulphur atoms in sulphides such as methyl phenyl sulphide and diphenyl sulphide (values estimated by molecular orbital calculations) are 5.91 and 5.86, respectively [40]. This density is lower for the thiophene derivatives (thiophene, benzothiophene, dibenzothiophene) because the S-electron pair participates in the aromatic delocalization. It varies from 5.69 to 5.76, and consequently the sulfur atom in thiophene derivatives is less nucleophilic. Similarly, due to the decreased nucleophilicity prompted by conjugation, the allylic and vinylic sulfides are less reactive than the dialkyl sulfide [41]. In brief, the order of reactivity of organic sulfur compounds depends on the nature of the radicals attached to the sulfur atom: dialkylsulfides > alkyl-arylsulfides > diarylsulfides > thiophenes [42–44].

The experimental results confirmed these theoretical estimations. Otsuki et al. [40] studied the relationship between the electron densities of sulfur atoms and reactivities for the oxidation of model sulfur compounds with hydrogen peroxide catalyzed by formic acid. They found a good correlation between the apparent rate constant and the electron density of the molecules. Our group found a similar ranking for the reactivity of thioethers and thiophene derivatives in the process of oxidation with $H_2O_2$ over solid catalysts, including Ti-containing molecular sieves and W-based HDL (Figure 1) [26,45,46].

**Figure 1.** Conversion of various sulfur compounds over Ti-SBA-15 catalyst; (o) THT; (Δ) MPS; (◊) PS; (□) DBT; Solvent = acetonitrile; T = 40 °C for THT and MPS; T = 60 °C for PS and DBT. Reprinted with permission from ref. [46]. Copyright 2018 Springer Nature.

Hulea et al. [26] showed that thiophenes are oxidized at different rates depending on their structure and the radical attached to the aromatic ring. Thus, the condensed aromatic thiophenes (BT and DBT) are more reactive than thiophene and substituted thiophenes. The oxidation of acetylthiophene occurs to a negligible extent because the acetyl group is an electron-withdrawing substituent. In view of industrial applications, it is important to note that the reactivity of aromatic sulfur compounds in oxidation with $H_2O_2$ contrasts with that observed in the hydrodesulfurization process. For comparison, Table 2 reports the relative reaction rates for various thiophene derivatives in oxidation and HDS reactions. It appears that sulfur compounds such as BT and DBT, which are hard to treat with HDS, are readily oxidized with $H_2O_2$ under mild reaction conditions.

**Table 2.** Sulfoxidation and HDS reactivity of thiophenes [26].

| Method | Conditions | Relative Rate Constants for Thiophene Desulfurisation Th:BT:DBT |
|---|---|---|
| Oxidation with $H_2O_2$ | Ti-Beta; MeCN; 60 °C | 1:5:3.5 |
| HDS | CoMo/$Al_2O_3$; 300 °C; 7–10 MPa | 22.5:13.3:1 |

The experimental data showed that the kinetic profile in the oxidation of sulfides is different from that observed for thiophenes. The oxidation of sulfides with $H_2O_2$ occurs in successive stages, leading to the corresponding sulfoxides (1-oxides) and sulfones (1,1-dioxides) (Scheme 2). The ratio between the sulfoxide and sulfone depends on the reaction time (Figure 2).

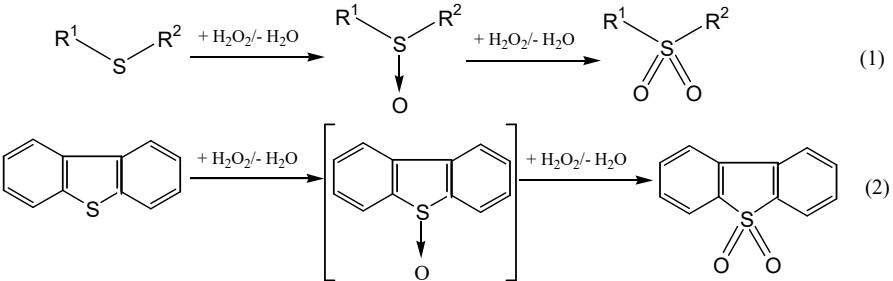

**Scheme 2.** Reaction pathways for the oxidation of sulfides (**1**) and thiophenes (**2**).

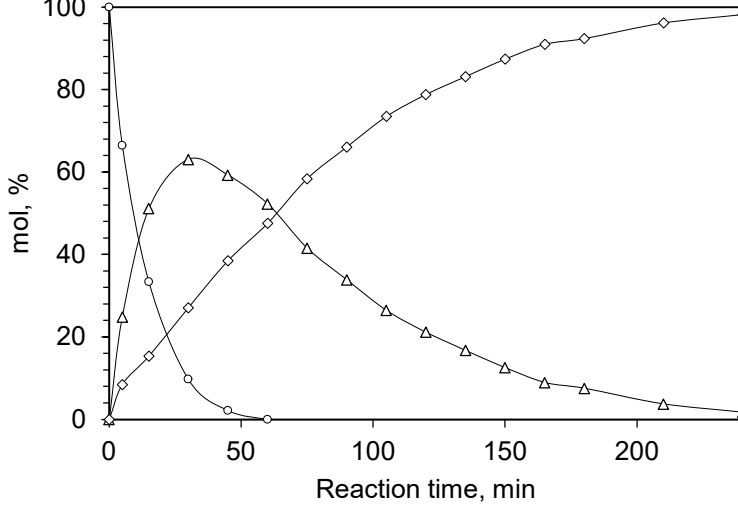

**Figure 2.** MPS oxidation with $H_2O_2$: molar composition of (o) MPS, (Δ) MPSO and (◊) MPSO2 vs. reaction time. Reprinted with permission from ref. [46]. Copyright 2018 Springer Nature.

In contrast, when the thiophene derivatives were oxidized, large amounts of corresponding sulfones were formed even at the first stage of the reaction [26,42]. Only traces of sulfoxides were produced, especially at lower temperature.

In some cases, reactions other than sulfoxidation can occur. For example, in the case of thiophenes that do not contain substituents at the aromatic ring, the oxidation to the S atom is strongly rivaled by a Diels–Alder reaction [26]. More precisely, thiophene 1-oxide and 1,1-dioxide undergo facile cycloaddition, in which one molecule acts as a diene and another as a dienophile, to produce sesquioxides (Scheme 3).

**Scheme 3.** Diels–Alder reactions involving sulfoxides and sulfones. Reprinted with permission from ref. [26]. Copyright 2001 Elsevier.

Recently, Bregante et al. [47] studied the oxidation of 2,5-dimethylthiophene with $H_2O_2$ in the presence of a Ti-Beta catalyst. They observed that part of 2,5-dimethylthiophene dioxide (sulfone) was converted into the corresponding epoxide by the oxidation of the C = C bonds within the thiophenic ring by $H_2O_2$.

### 3. Catalysts for Sulfoxidation with $H_2O_2$

The oxidation of thioethers with $H_2O_2$ in the presence of homogeneous catalytic systems based on transition metals (Ti, Mo, Fe, V, W, Re, Ru and Mn) is a reaction that has been known for a long time [21,22,48,49]. Thiophene derivatives can also be oxidized with $H_2O_2$ using efficient homogeneous catalysts such as HCOOH, $CCl_xCOOH$ ($x$ = 1–3), $CF_3COOH$ [50], $CF_3COOH$ [51] methyltrioxorhenium(VII) [52], and phosphotungstic acid [25]. Reddy et al. [53] reported the use of a heterogeneous catalyst in sulfoxidation reactions for the first time in 1992. They showed that the thioethers can be oxidized with hydrogen peroxide in the presence of titanosilicalites such as TS-1 and TS-2. Hulea et al. [26] showed for the first time that the thiophene derivatives are efficiently oxidized with $H_2O_2$ using Ti-silicates such as Ti-Beta and Ti-hexagonal mesoporous silica (Ti-HMS). Since then, a large variety of materials, consisting of transition metals incorporated in microporous, mesoporous and lamellar matrixes, have been investigated as catalysts for the oxidation of organic sulfur compounds [17,28,54,55]. Table 3 summarizes the main catalysts and the typical conditions in sulfoxidation reactions with $H_2O_2$.

**Table 3.** Catalytic sulfoxidation with $H_2O_2$: main catalytic systems and reaction conditions.

| Catalyst | Matrix | Substrate [a] | T (°C)/Solvent [b] | Ref |
|---|---|---|---|---|
| **Titanosilicates** | | | | |
| TS-1, TS-2 | Microporous | Thioethers | 56/MeCOMe | [53] |
| TS-2 | Microporous | Thioethers | 60/MeCN | [56] |
| TS-1, Ti-Beta | Microporous | Thioethers Sulfoxides Thiophenes | 30–60/MeOH, EtOH, MeCN, *t*-BuOH, MeCOMe, THF | [57–60] |

| Ti-beta | Microporous | MPS, phenyl-iso-pentyl | 40/MeCN | [61] |
| Ti-MCM-41 | Mesoporous | sulphide | | |
| Ti-Beta | Microporous | Thioethers | 20/MeCOMe | [62] |
| TS-1, Ti-Beta | Microporous | Thioethers | 21/MeOH | [63] |
| TS-1, Ti-Beta, Ti-HMS | Microporous Mesoporous | PS, Th, BT, DBT, THT | 60/MeOH, MeCN, EtOH, *t*-BuOH | [26] |
| Ti-MMM | Mesoporous | MPS | 20/MeCN | [64] |
| Ti-SBA-15 | Mesoporous | MPS | 20–50/MeCN, MeOH, MeCOMe | [65] |
| TS-1 | Microporous | Thiophene | 60/*n*-octane | [66,67] |
| Ag/TS-1 | Micro-, mesoporous | Thiophenes | 60/*n*-octane | [68] |
| TS-1 | Microporous | Thiophenes | 60/MeOH | [69] |
| Ti-Beta | Microporous | Thiophenes | 80/heptane | [70] |
| Ti-HMS and Ti-MSU | Mesoporous | DBT | 60/*n*-octane | [71] |
| Ti-MCM-48 | Mesoporous | DMSO, DBT, MPS | 30–50/MeCN | [72] |
| Ti-SBA-16 | Mesoporous | DBT | 60/MeCN | [73] |
| Ti-SBA-15 | Mesoporous | THT, MS, MPS, DBT | 40–70/MeCN | [46] |
| Ti-Beta | Microporous | 2,5-DMT | 40/MeOH, MeCN, EtOH, MeCOMe, DMSO, Dioxane | [47] |
| meso-TS-1 | Mesoporous | Th, DBT | 60/*n*-octane | [74] |
| meso-TS-1 | Mesoporous | Th, BT | 60/*n*-octane | [75] |
| meso-TS-1 | Mesoporous | Th, BT | 60/MeOH | [76] |
| Ti-HMS | Mesoporous | 4,6-DMDBT, BT, DBT | 60/MeOH | [77] |
| TS-1, Ti-Beta, Ti-MCM-41, Ti-MWW | Microporous Mesoporous | BT, DBT, 4,6-DMDBT | 30–90/MeCN | [78] |
| Meso-Ti-MOR | Mesoporous | DBT, 4,6-DMDBT | 60/MeCN | [79] |
| meso-TS-1 | Mesoporous | Th, BT, DBT, 4,6-DMDBT | 60/*n*-octane | [80] |
| Ti/SiO$_2$ | Mesoporous | DBT, 4,6-DMDBT | 60/*iso*-octane | [32] |
| Ti/SiO$_2$ | Mesoporous | BT, DBT, 4,6-DMDBT | 25–65/MeCN | [81] |
| meso-TS-1 | Mesoporous | DBT, 4,6-DMDBT | 60/octane | [82] |
| Ti-HMS/TS-1 | Micro-mesoporous | Th, BT, DBT | 60/MeOH | [83] |
| TiO$_2$-SiO$_2$ | Mesoporous | DPS, DBT, 4,6-DMDBT, DMSO, THT, MPS, BT | 20–70/MeCN | [34,84–86] |
| TiO$_2$-SiO$_2$ | Mesoporous | MPS, DBT | 60/MeCN | [87,88] |
| **Me-silicates** | | | | |
| VS-2 | Microporous | Thioethers | 60/MeCN | [89] |
| V-, Mn-zeolite | Microporous | DBT | 55/oil | [90] |
| Nb-Beta, Ta-Beta, Zr-Beta | Microporous | 2,5-DMT | 40/MeOH, MeCN, EtOH, MeCOMe, DMSO, Dioxane | [47] |
| Nb/SiO$_2$ | Mesoporous | MPS, BT, DBT, 4,6-DMDBT | 25–65/MeCN | [81] |
| VS-1, VS-2 | Microporous | BT, DBT | 60/MeCN | [91] |
| V-HMS | Mesoporous | | | |
| Mo-TS-1 | Microporous | Thioethers | 20/MeOH | [92] |
| V-SBA-15 | Mesoporous | MPS | 20–50/MeCN, MeOH, MeCOMe | [65] |
| **Me-MOF** | | | | |
| Sc-MOF | Microporous | Thioethers | 50/MeCN | [93] |
| Zn-MOF | Microporous | Thioethers | 25/MeCN, CH$_2$Cl$_2$ | [94] |

| | | | | |
|---|---|---|---|---|
| La-MOF | Microporous | MPS | 60/MeCN | [95] |
| Yb-LRH, Yb-RPF-5 | Microporous | MPS | 60/MeCN | [96] |
| Sc-MOF, Y-MOF | Microporous | Thioethers | 60/MeCN | [97] |
| Cr-MIL-101 | Microporous | Aryl sulfides | 25/MeCN | [98] |
| Zr- UiO-66 | Microporous | DBT, 4-MDBT, 4,6-DMDBT | 50/MeCN, octane | [99] |
| Zr- UiO-66 | Microporous | DBT | 30–70/octane | [100] |
| Ti-Zr-UiO-66 | Microporous | DBT | 60/*n*-octane, MeCN | [101] |
| Ti-MOF (MIL-125) | Microporous | Th, BT, DBT, 4,6-DMBT | 60/*n*-octane | [102] |
| UiO-66 | Microporous | DBT | 60/MeCN | [103] |
| nitro-/amino- UiO-66(Zr) | Microporous | DBT, 4,6-DMDBT | 60/*n*-octane | [104] |
| Zr- UiO-66 | Microporous | BT, DBT | 50/*n*-octane | [105] |
| Zr- UiO-66, UiO-67, NU-1000, and MOF-808 | Microporous | DBT | 40/MeCN | [106] |
| Ti-Zr-UiO-66 | Microporous | MPS, DBT | 60/MeCN | [107] |
| Zr- UiO-66 | Microporous | MPS | MeOH, MeCN, CH$_2$Cl$_2$ | [108] |
| UiO-66 and UiO-67 | Microporous | MPS, MPSO | 27/MeCN | [109] |
| Zr-abtc and MIP-200 | Microporous | MPS, MPSO, Thianthrene | 27/MeCN | [110] |
| **Me-LDH** | | | | |
| W-LDH | Layered | | 25/H$_2$O$_2$ | [41] |
| W-LDH | Layered | Thioethers BT, DBT | 40–70/MeCN, EtOH, MeCOMe, *n*-BuOH, *t*-BuOH | [45] |
| V, Mo, W-LDH | Layered | MPS, PPS, BPS, BT, DBT | 40/MeCN | [111] |
| V, Mo, W-LDH | Layered | THT | 20–30/Dioxane, MeCN, EtOH, MeOH, MeCOMe | [112] |
| V, Mo, W-LDH | Layered | MPS, DBT, DMSO | 30–50/MeCN, H$_2$O | [20] |
| W-LDH | Layered | DMSO | 25–50/H$_2$O, EtOH, Dioxane, MeCN | [33] |
| Mo-LDH | Layered | MPS, DBT | 40–70/MeCN | [113] |
| Mo-, V-LDH | Layered | DBT | 70/MeCN | [114] |

[a] Th = thiophene; BT = benzothiophene; DBT = dibenzothiophene; THT = tetrahydrothiophene; MPS = methyl-phenyl-sulfide; PS = phenyl sulfide; MPSO = methyl-phenyl-sulfoxide; DMSO = dimethylsulfoxide; 2,5-DMT = 2,5-dimethylthiophene; 4-MDBT = methyl-dibenzothiophene; 4,6-DMDBT = 4,6-dimethyl-dibenzothiophene. [b] MeOH = methanol; EtOH = ethanol; BuOH = butanol; THF = tetrahydrofuran; MeCOMe = acetone; MeCN = acetonitrile; DMSO = dimethylsulfoxide.

### 3.1. Ti-Containing Catalysts

Due to their unique catalytic potential in oxidation reactions with H$_2$O$_2$, titanium-containing silicas (titanosilicates) are the most-explored catalysts for both academic and industrial purposes. The creation of catalytic sites in these materials is attributed to the intimate interaction between TiO$_2$ and SiO$_2$ allowing the formation of Ti–O–Si bonds. The isolated titanium atom in a silica matrix, which is not connected to any other titanium atom via oxygen bridges, is considered to be responsible for the formation of peroxo species when it is in contact with H$_2$O$_2$. To establish the environment of Ti-sites in catalysts, spectroscopic techniques such as DR UV-vis, XPS, EXAFS/XANES, Raman and FT-IR can be used (see the review [115]). For example, in DR UV-vis spectra, the band at 210–230 nm (Figure 3) is attributed to an oxygen-to-metal charge transfer at the isolated tetrahedral Ti(IV) center [116,117]. Bands in the 230–320 nm region indicate the presence of isolated 5/6-coordinated Ti atoms or of small TiO$_2$ nano-domains [118]. Note that these last species are responsible for the direct decomposition of H$_2$O$_2$ into inactive molecular oxygen and water [119]. In other words, the H$_2$O$_2$ efficiency in sulfoxidation is directly related to the number of isolated Ti sites.

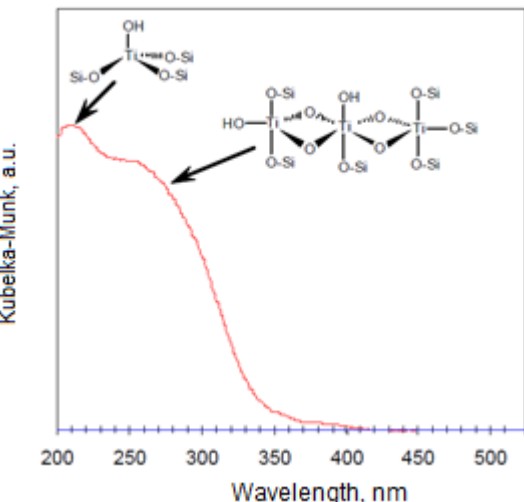

**Figure 3.** Diffuse reflectance UV-visible spectrum of $TiO_2$–$SiO_2$. Previously unpublished figure.

TS-1 is the original titanosilicate [5,6]. This highly stable microporous material has an exceptional ability to catalyze the oxidation of many organic molecules, including sulfoxidation with aqueous $H_2O_2$. [7–11]. However, since the Ti active sites are located inside of a channel system of about a 0.55 nm average diameter, the use of TS-1 is restricted to reactions of relatively small molecules [26,71]. Due to the TS-1 pore size/architecture and diffusion limitations, *shape selectivity* effects were observed in the reactions catalyzed by this material. For sulfoxidation reactions with $H_2O_2$ on TS-1, we can distinguish three types of shape selectivities:

(i) reactant selectivity (when the pore size limits the entrance of the reacting molecules). For example, the activity of TS-1 is very high for the oxidation of MPS (a small molecule which can penetrate the pores), while it is negligible when larger molecules, such as dibutylsulfoxide [60] or DBT [26,86], are oxidized.

(ii) product selectivity (when the pore size limits the desorption of the product molecules). Robinson et al. [63] studied the shape-selective oxidation of TS-1 using different isomeric butyl methyl thioethers. Based on the experimental results and calculated occupying volumes, the authors suggested that in the case of two molecules, namely *tert*-butyl and *iso*-butyl methyl sulfides, the shape selectivity inhibited the formation of the bulkier sulfones.

(iii) transition state selectivity (when the pore size avoids the formation of certain intermediates). Adam et al. [120] studied the sulfoxidation of thianthrene 5-oxide (T-5-O) with $H_2O_2$ over TS-1, Ti-Beta and Ti-MCM-41. Only TS-1 was inactive in sulfoxidation. This behavior was attributed to an inhibition caused by the size of the transition state involved in reaction.

To overcome the limitation exhibited by TS-1, several research groups have made efforts in the design of new microporous and mesoporous titanosilicates with enhanced accessibility to the active centers [121]. First, the Ti-microporous crystalline silicates, such as Ti-Beta (pore size of 0.76 × 0.64 nm) [26,61] and Ti-MWW (with supercages of 0.7 × 0.7 × 1.8 nm) [78], were effective catalysts for the sulfoxidation reactions of some bulky substrates.

Ti-containing mesoporous templated silica, such as Ti-HMS, Ti-MCM-41, Ti-MCM-48, Ti-SBA-15, were also successfully used in the sulfoxidation reaction of large molecules (BT, DBT, 2,4-DMDBT) with $H_2O_2$ as oxidizing agent (see Table 3). Unfortunately, these materials usually exhibited lower catalytic performance (in terms of activity and $H_2O_2$ efficiency) than in the oxidation of organic substrates with aqueous solutions of $H_2O_2$.

Consequently, their use was limited to oxidation reactions with dry organic hydroperoxides. This behavior has been ascribed to the much higher hydrophilicity of the amorphous materials leading to extensive water adsorption, hindering the access to the Ti(IV) active sites [122,123], and also to the irreversible deactivation of these sites in the presence of aqueous $H_2O_2$ [124,125]. Additionally, these materials showed a low hydrothermal stability.

Hulea et al. [26] investigated the behavior of some catalysts having different topology and porosity. They compared three catalysts, i.e., TS-1 (pore size of 0.55 nm), Ti-Beta (pore size of 0.76 × 0.64 nm) and Ti-HMS (pore size of 4 nm), in sulfoxidation with $H_2O_2$ of 2,5-DMT, benzothiophene (BT) and dibenzothiophene (DBT). The catalysts contained nearly equivalent amounts of framework Ti. Figure 4 shows the conversions of thiophene derivatives over the three catalysts, obtained under similar conditions.

For the small molecule 2,5-DMT, the order of catalyst activity was TS-1 > Ti-Beta > Ti-HMS, suggesting a higher intrinsic activity of the Ti atoms in TS-1 than in Ti-Beta and Ti-HMS. The absence of any activity in the sulfoxidation of BT and DBT over TS-1 indicates clearly that these molecules hardly penetrate the TS-1 micropores, contrasting with the larger-pore Ti-Beta and mesoporous Ti-HMS. In the case of the sulfoxidation of BT, which can penetrate the pores of Ti-Beta, the activity is higher than with the Ti-mesoporous material, Ti-HMS. On the contrary, when a larger molecule such as DBT was oxidized, Ti-Beta proved slightly less active that Ti-HMS, suggesting that DBT diffusion is limited in the 12-member ring pores of Ti-Beta zeolite.

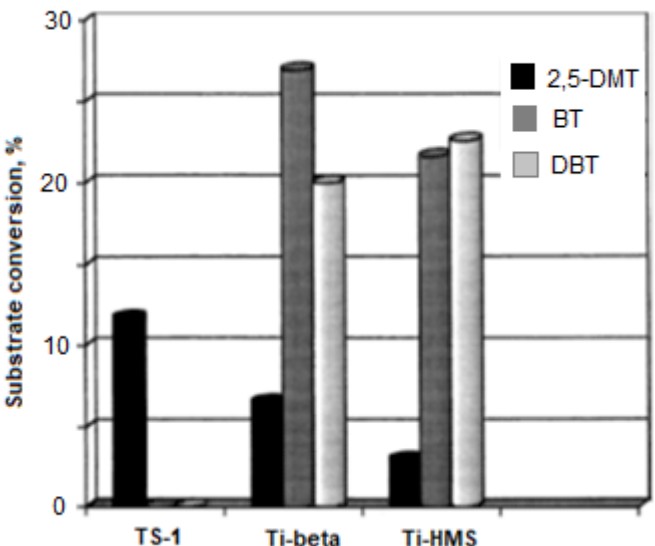

**Figure 4.** Influence of catalyst type on the conversion of thiophene derivatives in the oxidation with $H_2O_2$. Reprinted with permission from ref. [26]. Copyright 2001 Elsevier.

Another approach used to improve the diffusion performance towards bulky molecules consisted of synthesizing zeolites with a hierarchical porous structure. By using templating processes, mesoporous TS-1 [76,82] and Ti-MOR [79] were prepared and evaluated as catalysts in the sulfoxidation reaction. These materials with highly accessible Ti sites demonstrated higher catalytic activities compared to conventional Ti-silicates in the oxidation of thiophene derivatives with large molecules, as shown in Figure 5.

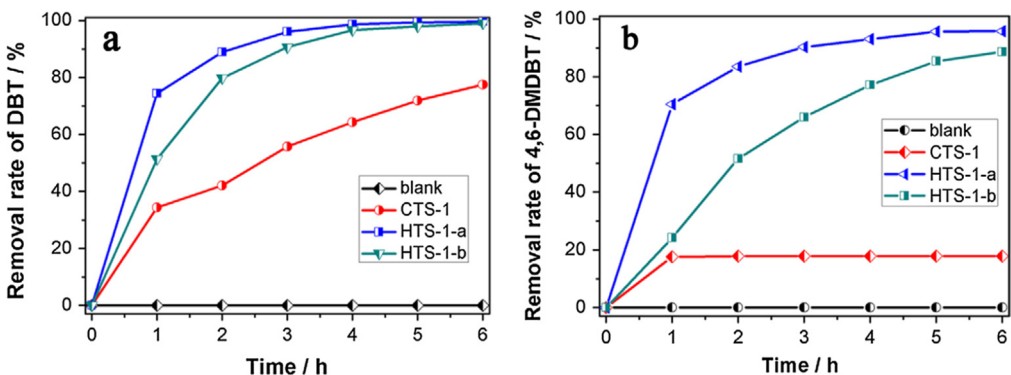

**Figure 5.** Oxidation of DBT (**a**) and 4,6-DMDBT (**b**) over conventional TS-1 (CTS-1) and mesoporous TS-1 (HTS-1-a and HTS-1-b). Reprinted with permission from ref. [82]. Copyright 2017 Elsevier.

### 3.2. Me-Silicates

In the open literature, we find several sulfoxidation studies investigating the behavior of Me-silicates as catalysts (Me = V, Mo, Nb, Ta, Zr). As shown in Table 2, both microporous and mesoporous materials were studied. The active species proposed for these catalysts are similar to those identified for the Ti-silicates (*vide infra*). Compared to Ti-silicates, Me-titanosilicates exhibited lower performances. Thus, in terms of sulfoxidation activity, Ti-Beta was much more active than Nb-Beta, Ta-Beta and Zr-Beta [47]. On the other hand, V-containing mesoporous catalysts showed very low stability in the reaction medium [65,91].

### 3.3. Me-MOF

Metal–organic frameworks (MOFs) are a class of materials composed of metal ions/clusters connected by multidentate organic linkers into a regular (micro)porous structure. They exhibit high surface areas and pore volume, tunable functionality and good accessibility to metal species. MOFs have shown encouraging behavior in applications such as gas storage, gas separation, drug delivery and heterogeneous catalysis [126–128].

Among the reactions catalyzed by MOFs, the selective oxidation of organic compounds with molecular oxygen, alkyl hydoperoxides and $H_2O_2$ has been widely investigated [129–132]. The oxidation of thioethers and thiophenes with $H_2O_2$ into corresponding sulfoxides and sulfones is the most investigated oxidation reaction catalyzed by MOFs [17]. The active oxidation catalytic sites of MOFs are the metal-connecting nodes with unsaturated coordination environments [128]. Most MOFs are thermally and hydrolytically less stable than zeolites and related materials. Exceptions are $Zr^{IV}$ terephthalate UiO-66 and Ti-MIL-125, which are leaching-tolerant and recyclable catalysts for $H_2O_2$-based oxidative transformations, including the sulfoxidation reactions. MOFs are microporous materials. As a result, the effect of steric hindrance has been observed in the case of Me-containing MOFs. Zhang et al. [102] evaluated the catalytic performance of Ti-MIL-125 in the oxidation with $H_2O_2$ of 4,6-DMDBT, DBT, BT and Th. The order of reactivity for the sulfur compounds was 4,6-DMDBT < Th < BT < DBT. Although the 0.8–1.22 nm molecular size of DBT is larger than the pores of MIL-125 (approximately 0.5–0.77 nm windows), this molecule was more reactive than the other compounds. In fact, the reaction mainly takes place on the outer surfaces of MIL-125, which contain high levels of titanium.

Generally, the nature of the metal in the MOF nodes determines the sulfoxidation mechanism, and thus the product selectivity [17]. While Zn-, Cr- and Ti-MOFs realize electrophilic activation of $H_2O_2$ and produce mainly sulfoxides, Zr-MOFs realize nucleophilic activation of $H_2O_2$ and produce mostly sulfones. This behavior of Zr-MOFs has been related to the presence of weak basic sites in their structure.

### 3.4. Me-LDH

Layered double hydroxides (LDHs) are anionic clays consisting of brucite-like sheets stacked in a layered structure, having counter-anions intercalated in the inter-layer space. Owing to their unique anion exchange ability, LDHs are attractive hosts for accommodating various organic and inorganic species in their interlayer space. This capacity has also been used for developing materials and applications in heterogeneous catalysis. Choudary et al. [41] used for the first time tungstate-exchanged Mg-Al-LDHs as catalysts for the oxidation of organic sulfides with hydrogen peroxide. In a series of studies, Hulea and coworkers extensively investigated the sulfoxidation catalytic potential of Mo-, W- and V-oxoanions intercalated in LDHs [33,45,111–114]. The catalysts were prepared by direct ion exchange of the anions originally existing in the interlayer of a LDH (e.g., $NO_3^-$) with metal-oxoanions, i.e., $WO_4^{2-}$, $W_7O_{24}^{6-}$, $V_2O_7^{4-}$, $V_{10}O_{28}^{6-}$, $MoO_4^{2-}$ and $Mo_7O_{24}^{6-}$. Figure 6 shows schematic representations of the intercalation of Mo/V-oxoanions. The gallery height increased from 8.7–8.9 Å (typical for the LDH containing nitrate anions) to 10.5 Å and 10.8 Å after exchange with molybdate and vanadate, respectively.

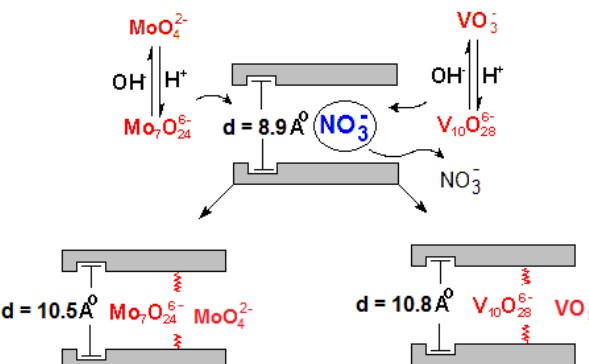

**Figure 6.** Schematic representation of the intercalation of Mo- and V-oxoanions by anion exchange. Previously unpublished figure.

For evaluating the modifications in gallery size and the extent of the anionic exchange, XRD is a very efficient method. Figure 7 shows the XRD patterns of parent and modified LDHs with Mo-oxoanions. The parent $NO_3$-LDH exhibited a XRD pattern characteristic of the well-crystallized layered structure, with typical $d_{003}$-values of 8.9 Å. The XRD patterns of Mo-LDHs displayed the same reflections characteristic of the layered materials, but at smaller values of 2θ. Changes in the $d_{003}$-value indicate an increase in the basal spacing and prove the replacement of smaller nitrate anions by larger anion species.

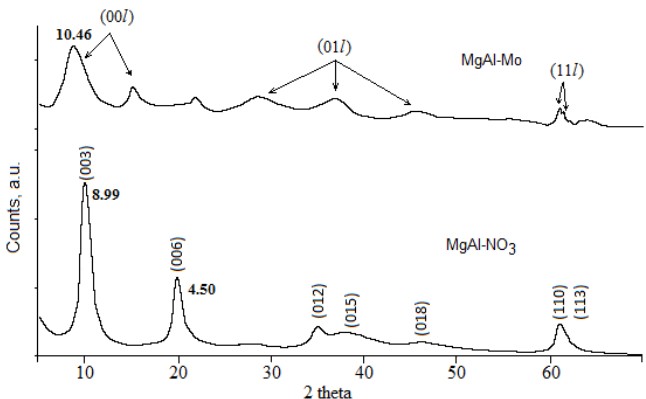

**Figure 7.** XRD patterns of Mo- and $NO_3$-MgAl-LDH. Adapted from ref. [113], used with permission. Copyright 2012 Elsevier.

Knowing the nature of the species inserted in the LDH structure is crucial for the oxidation reaction. The spectral techniques are usually used for this purpose. For example, the Raman spectroscopy was successfully used for the characterization of LDHs containing V-, W- and Mo-based oxoanions in the interlayer space [113,114]. This method provided precise information regarding the type of the oxoanions, as shown in Figure 8. In the spectrum of the parent LDH-NO$_3$ hydrotalcite, the intense band at 1050 cm$^{-1}$ is assigned to the nitrate vibrations. After exchange with Mo-oxoanions, this band disappeared and new bands are evidenced at: 895 cm$^{-1}$ ($MoO_4^{2-}$), 920 cm$^{-1}$ ($Mo_2O_7^{2-}$) and 947 cm$^{-1}$ ($Mo_7O_{24}^{6-}$).

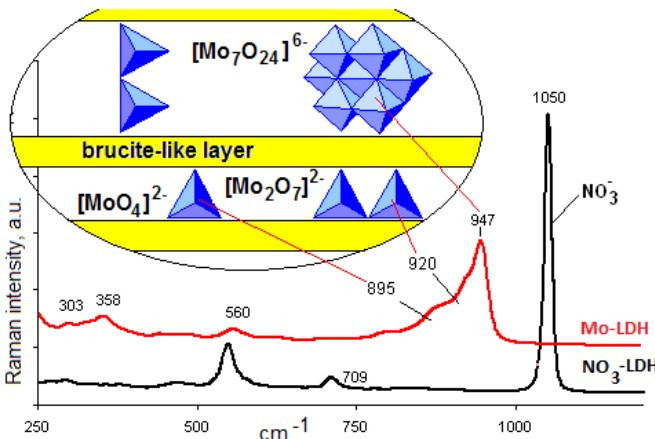

**Figure 8.** Raman spectra of NO$_3$-LDH and Mo-LDH. Reprinted with permission from ref. [113]. Copyright 2012 Elsevier.

## 4. Active Species and Mechanism in the Oxidation with H$_2$O$_2$

### 4.1. Me-Silicates

The role of the metal (Ti, W, V, Mo, etc.) contained in the oxidation catalysts is to activate H$_2$O$_2$. It is widely accepted that the species generated at Me sites upon H$_2$O$_2$ contact are Me-OOH (Me-hydroperoxo) and Me-(O$_2$) (Me-peroxo), which are valuable intermediates for the oxidative processes. (Scheme 4).

**Scheme 4.** Formation of the Ti-(hydro)peroxo site. Adapted from ref. [86], used with permission. Copyright 2010 Elsevier.

UV-visible spectroscopy gives direct evidence of the formation of these species. For example, Zecchina et al. [133] found that, in contact with H$_2$O$_2$, the Ti-silicates turn yellow and exhibit a specific absorption band at 26,000 cm$^{-1}$ (385 nm). In a sulfoxidation study, Cojocariu et al. [86] prepared TiO$_2$–SiO$_2$ xerogels using a non-hydrolytic sol-gel method. These materials exhibited unique textures (specific surface area 1200 m$^2$ g$^{-1}$, pore volume

2.4 cm$^3$ g$^{-1}$, average pore diameter 15 nm) and excellent performance in the mild oxidation of bulky sulfides and thiophenes with aqueous solutions of $H_2O_2$. The authors investigated the nature of the Ti species resulting from the interaction of the catalyst with $H_2O_2$. The DR UV-vis spectra of $TiO_2$-$SiO_2$/$H_2O_2$, subjected to various conditions of treatment, were recorded and illustrated in Figure 9. The spectrum of untreated $SiO_2$-$TiO_2$ (spectrum a) exhibited two main absorption bands at 210 and 260 nm. These bands are assigned to Ti-isolated sites and oligomeric Ti species, respectively. The treatment of $TiO_2$-$SiO_2$ with $H_2O_2$ produced a rapid change in the catalyst color from white to pale yellow and UV spectra showed the apparition of an additional broad band at 250–400 nm (spectrum b). This band is usually attributed to bidentate ($\eta^2$) titanium hydroperoxo species [134].

After drying at 550 °C (spectrum c), the original adsorption feature was almost completely restored. These spectroscopic data indicate that (i) titanium hydroperoxo catalytic species are generated by cleavage of Ti–O–Si bridges in the presence of $H_2O_2$ and (ii) the cleavage of Ti–O–Si bonds in these catalysts is largely reversible.

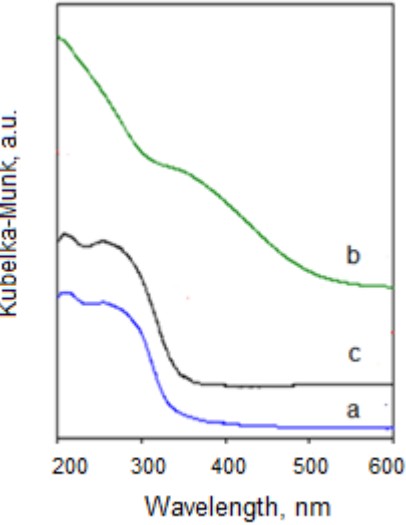

**Figure 9.** DR UV-vis spectra of $TiO_2$-$SiO_2$: calcined (curve a); after adsorption of $H_2O_2$ (curve b); then followed by thermal activation in air at 550 °C (curve c). Adapted from ref. [86], used with permission. Copyright 2010 Elsevier.

Using in situ UV-vis spectroscopy, Bregante et al. [47] showed that, besides Ti, metals such as Zr, Nb and Ta substituted in a silicate framework irreversibly activate $H_2O_2$ to form intermediate metal-hydroperoxides (Me-OOH) and peroxides (Me-(O$_2$)).

Spectroscopic approaches other than UV-vis have been used to investigate the Me-(hydro)peroxo species formed upon dosing $H_2O_2$ on Ti-silicates. Using in situ FT-IR spectroscopy, Lin and Frei [135] reported the first direct observation of the active Ti-OOH species formed upon the interaction of Ti-silicate with $H_2O_2$. Geobaldo et al. [136] used the EPR method. Bonino et al. [124] investigated the structure of the complexes formed between $H_2O_2$ and TS-1 by EXAFS spectroscopy. The EXAFS results, supported by acidity measurements and titration of active oxygen content, showed that the O–O species, responsible for the yellow color of the catalyst, is a side-on peroxo complex, probably generated by the reversible rupture of one Ti–O–Si bridge, with the formation of Ti(O$_2$H) and H–O–Si groups.

Tozzola et al. [137] carried out a detailed vibrational and computational study in order to describe the active sites formed over Ti-silicate. The vibrational IR and Raman experiments suggested that under neutral conditions, in the presence of $H_2O_2$, an unstable Ti complex containing the O–O moiety (plausibly an OOH group) was formed. This species was transformed in basic conditions into a more stable peroxo form, absorbing at 840

cm$^{-1}$ (Raman) and 836 cm$^{-1}$ (IR). The assignment of the spectra was confirmed by ab initio calculations and by parallel experiments performed on the structurally similar Ti-free silicalite.

A typical catalytic cycle for sulfoxidation of thioethers and thiophenes over Me-silicates is comparable to that proposed by Bregante et al. [47] for the sulfoxidation of methylthiophene ($C_6H_8S$, Scheme 5). The catalytic cycle involves the quasi-equilibrated adsorption of $C_6H_8S$ (step 1) and $H_2O_2$ (step 2), followed by the irreversible activation of $H_2O_2$ (step 3) to form a pool of M-OOH and M-($\eta^2$-$O_2$) intermediates (denoted as M-($O_2$)). The active M-($O_2$) species react with $C_6H_8S$ (step 4) to form the corresponding sulfoxide, which then desorbs (steps 6).

**Scheme 5.** Proposed mechanism for the oxidation of $C_6H_8S$ and $C_6H_8SO$ with $H_2O_2$ over M-Beta (M = Ti, Zr). Adapted from ref. [47], used with permission. Copyright 2018 Elsevier.

*4.2. Me-LDH*

Maciuca et al. [111] investigated the sulfoxidation reaction with $H_2O_2$ using W-LDH catalysts. Direct evidence of the formation of peroxotungstate species when the catalyst was in contact with $H_2O_2$ was provided by UV-vis spectroscopy. Thus, the $\lambda_{max}$ shifted from 250 nm (in dry W-LDH) to 325 nm (in $H_2O_2$-W-LDH), confirming the formation of peroxotungstate intermediates (Figure 10).

A plausible catalytic cycle for the oxidation of sulfides to sulfoxides, which is the first stage of the oxidation process, over a W-containing LDH catalyst, is depicted in Scheme 6.

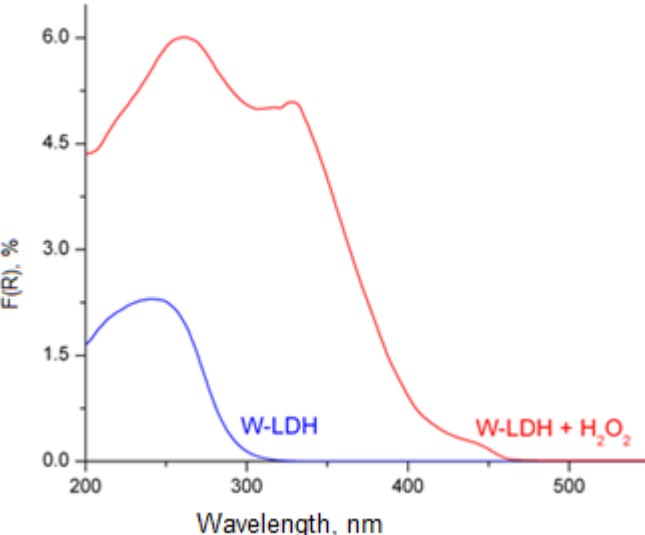

**Figure 10.** UV-vis -DR spectra of W-LDH before and during the oxidation reaction with $H_2O_2$. Reprinted with permission from ref. [111]. Copyright 2008 Elsevier.

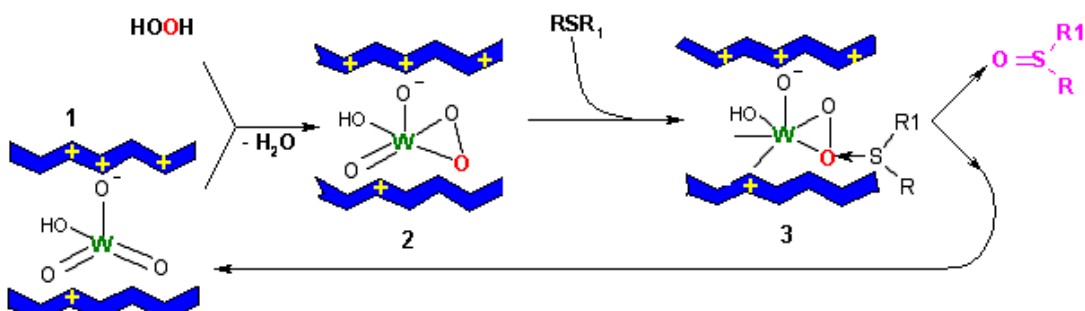

**Scheme 6.** Proposed mechanism for the sulfide oxidation with $H_2O_2$ over W-LDH catalyst. Reprinted with permission from ref. [111]. Copyright 2008 Elsevier.

### 4.3. Me-MOFs

When Me-MOFs were used as catalysts for sulfoxidation with $H_2O_2$, Me-OOH species similar to those observed with other catalysts were identified. Experimental and computational results published by Limvorapitux et al. [108] revealed that the $Zr_6$-oxo-hydroxo nodes of UiO-66 (Zr-$\mu_1$-OH) became Zr-$\mu_1$-OOH after reacting with $H_2O_2$. The Zr-$\mu_1$-OOH species are active in oxidizing the sulfide as well as its sulfoxide product.

In a recent work, Zalomaeva et al. [109] investigated thioether sulfoxidation with $H_2O_2$ over Zr-MOF. They used Raman spectroscopy to probe the interaction of Zr-MOF (Zr-UiO-66) with aqueous hydrogen peroxide in an MeCN medium. As shown in Figure 11, two new Raman features, at 834 and 1124 cm$^{-1}$ appeared when $H_2O_2$ had been added to the MOF catalyst (spectra B and C). The band at 834 cm$^{-1}$ was assigned to the O–O stretching mode of a peroxide ligand. The intensity of this band markedly increased when the sample was not washed with MeCN after the treatment (spectrum D). The authors consider that the second intense band at 1124 cm$^{-1}$ indicates the formation of a superoxide complex.

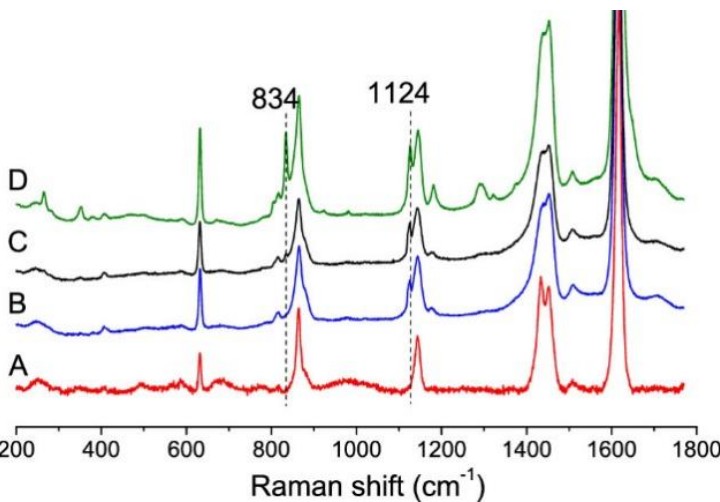

**Figure 11.** Raman spectra of fresh UiO-66 (A) and $H_2O_2$-treated UiO-66: (B and C) 30- and 90-fold excess of $H_2O_2$ followed by washing with MeCN; (D) 60-fold excess of $H_2O_2$ without washing. Reprinted with permission from ref. [109]. Copyright 2020 American Chemical Society.

By analyzing the oxidation products, as well as by using the kinetic and spectroscopic data, several structures of peroxo zirconium species were proposed (Scheme 7). These species are considered responsible for the nucleophilic oxygen transfer in Zr-MOF-catalyzed oxidations.

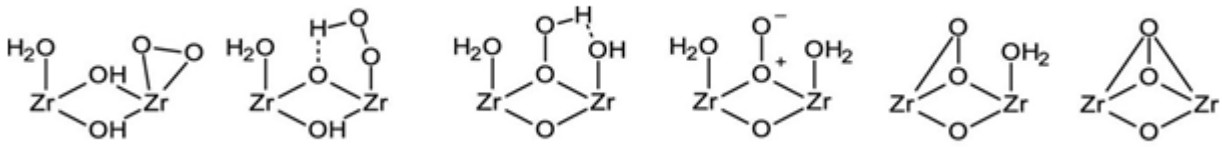

**Scheme 7.** Conceivable structures of nucleophilic peroxo zirconium species. Reprinted with permission from ref. [109]. Copyright 2020 American Chemical Society.

Zheng et al. [106] studied the DBT sulfoxidation with $H_2O_2$ over Zr-MOFS. Based on kinetic and ESR tests, the authors demonstrated that both $\bullet O_2^-$ and $\bullet OH$ radicals (derived from $H_2O_2$ decomposition) are present in the DBT oxidation. The catalytic activities of Zr-MOFs have been related to the Brönsted and Lewis acidity of the catalyst. A mechanism including seven steps was proposed (Scheme 8): (a) protonation of Zr–OH sites; (b) dehydration of Zr–OH$_2^+$ species ; (c, d) the formed unsaturated Zr-sites act as Lewis acid reacting with $H_2O_2$ to form Zr–OOH catalytic species; (e) upon heating, the O−O bond cleavage in the Zr–OOH to form $\bullet OH$ radicals; (f) Zr–O$\bullet$ reacts with $H_2O_2$ to generate $\bullet O_2^-$; (g) $\bullet OH$ and $\bullet O_2^-$ oxidize DBT to form sulphone.

**Scheme 8.** Proposed mechanism of DBT oxidation with $H_2O_2$ over Zr-MOFs. For (**a**)–(**g**) see the main text. Adapted from ref. [106] and used with permission. Copyright 2019 American Chemical Society.

### 5. Understanding the Role of Solvents

Sulfoxidation reactions with $H_2O_2$ are carried out in the presence of protic or aprotic solvents, which form a single phase with the substrate and the hydrogen peroxide solution. It is known that the nature of the solvent plays a very important role in the catalytic reactions carried out in liquid phase, and the results obtained in the sulfoxidation of organic substrates with $H_2O_2$ over solid catalysts confirm this assertion [138]. The solvent has an important effect on the outcome of the reaction, i.e., on reaction kinetics, yields, and by-product formation. The early studies on the effect of the solvent in the oxidation reaction with $H_2O_2$ were performed on the epoxidation of lower olefins with TS-1 as the catalyst. Methanol or other small protic alcohols were the preferred solvents for these reactions [139]. The favourable role played by alcohols was explained in terms of the hindrance of adsorbed solvent molecules in the zeolite channels or by the electronic effect of the adsorbed solvent molecule on the catalytic active centers. Clerici and Ingallina [140] suggested that the alcohols are involved in the elementary steps, forming solvated titanium peroxo species (Scheme 9).

**Scheme 9.** Formation of solvated titanium peroxo species. Adapted from ref. [140], used with permission. Copyright 1993 Elsevier

In a detailed study, Hulea and Moreau [58] evaluated the role played by different solvents (lower alcohols, acetone, MeCN and tetrahydrofurane (THF)) in the oxidation of thioethers catalyzed by TS-1 and Ti-Beta. In TS-1, the order of efficiency of the solvents was: MeOH > EtOH > MeCN > *t*-BuOH > MeCOMe > THT. The activity order for ROH type solvents, MeOH > EtOH > *t*-BuOH, is explained based on increasing electrophilicity ($\varepsilon$, $pK_{auto}$, ref. [58]) and steric constraints of the intermediate species (see Scheme 9) inside the channels of TS-1. In the case of Ti-Beta, the behavior of the protic solvents was very

similar (MeOH ~ EtOH ~ *t*-BuOH) and much higher than those obtained in the aprotic solvent ROH > MeCN > MeCOMe > THT.

The same group studied the effect of the solvent on the oxidation of thiophenes catalyzed by TS-1, Ti-Beta and Ti-HMS [26]. The solvents used were acetonitrile, methanol, ethanol, and t-butanol. Acetonitrile, as an aprotic polar compound, showed higher efficiency with respect to the protic solvents such as MeOH, EtOH or *t*-BuOH. This behavior has been related to the hydrophilic/hydrophobic character and to the acid properties of the catalyst. Ti-Beta zeolites contain hydroxyl groups with acidic and hydrophilic characters [141]. The protic solvents (ROH), $H_2O_2$ and water (as a reaction product) can be adsorbed by this material and the inner substrate concentration is lower. In contrast to the hydrophilic character of Ti-Beta (and Ti-mesoporous materials), TS-1 has a hydrophobic character. For the later catalyst, methanol or other small protic compounds are the preferred solvents for oxidation reactions [64,142]. The positive effect of the MeCN in the oxidation reactions over the Ti-Beta catalyst was also associated with the poisoning of the acid sites by basic molecules of the solvent [143] or with a possible base-catalyzed reaction between $H_2O_2$ and the solvent, leading to peroxyimidic acid, $H_3C–C(=NH)–OOH$ [144–146]. This acid is known to be an active oxidizing agent for various organic substrates.

The experimental results obtained in sulfoxidation reactions catalyzed by Me-LDH are in agreement with those previously obtained over Ti-containing molecular sieves [45,112]. Thus, acetonitrile proved to have the best performance, but a favorable effect of the protic solvents (ethanol, *t*-butanol, *n*-butanol) was also observed. In the presence of acetone, only a small conversion of substrate was obtained.

Choudary et al. [41] related the solvent effect to the hydrophilicity of the LDH catalysts. Studying the oxidation of thioanisole over an LDH-WO₄ catalyst, they found the following order for the solvent efficiency: $H_2O \cong CH_3OH > CHCl_3 > CH_3CN > CH_2Cl_2$. The hydrophilicity of the LDH material makes the hosted oxidation catalyst water-compatible, and thus the best results among the solvents investigated were obtained in water. Bregante et al. [47] correlated the rates of methyl-thiophene ($C_6H_8S$) oxidation with the nucleophilicity of the solvent used. Over the Ti-Beta catalyst, the rates decreased in the order acetonitrile > *p*-dioxane ~ acetone > ethanol ~ methanol. In situ UV-vis spectra showed that highly nucleophilic solvent molecules competed effectively for active sites, inhibited $H_2O_2$ activation and formation of reactive M-OOH and M-($\eta^2$-O₂) species and gave lower turnover rates. As shown in Figure 12, thiophene consumption rates decreased exponentially with the solvent nucleophilicity (N1).

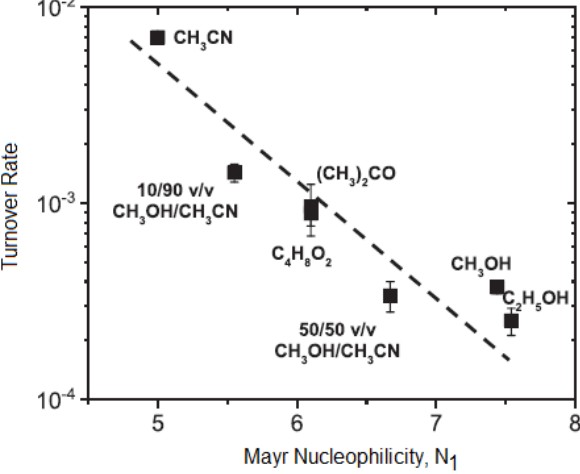

**Figure 12.** Turnover rates (mol $C_6H_8SO$)(mol M · s)$^{-1}$ for the oxidation of $C_6H_8S$ as a function of solvent nucleophilicity over Ti-Beta. Adapted from ref. [47], used with permission. Copyright 2018 Elsevier.

Limvorapitux et al. [108] studied the effect of the solvent on the product distribution obtained in a UiO-66 MOF catalyst. They found that the sulfoxidation reaction carried out in the $CH_3OH$ solvent led to higher selectivities for the sulfoxide product, while overoxidation to sulfone predominated in $CH_3CN$ and $CH_2Cl_2$ solvents. Kinetic studies and computational evaluations supported a model where the sulfoxide product can bind to a site adjacent to the active catalyst species in these latter solvents, resulting in higher degrees of overoxidation through increased local concentration. Such an effect was minimized in the $CH_3OH$ solvent, which can interact more strongly with the open sites on the nodes than sulfoxide does, and thus maintain good sulfoxide selectivity.

In short, most studies pointed out the excellent behavior of acetonitrile as a solvent in the sulfoxidation reactions. Lower alcohols, particularly methanol, can be efficient solvents for the sulfoxidation of thioethers.

### 6. Conclusions

The large number of studies reviewed above undoubtedly shows the high scientific interest of the sulfoxidation reaction carried out in the presence of $H_2O_2$ and heterogeneous catalysts. Focusing on the catalytic behavior of Me-based inorganic porous catalysts (Me = Ti, V, Mo, W, Zr), this review shows the high potential of these materials for the oxidation of thioethers and thiophenes under (very) mild conditions.

Some concluding remarks can be given after the above extensive discussion:

i. The order of reactivity of the organic compounds depends on type of the radicals attached to the sulfur atom: dialkylsulfides > alkyl-arylsulfides > diarylsulfides > thiophenes.
ii. Ti-based catalysts, particularly Ti-silicates, show superior catalytic behavior compared to that exhibited by the other Me-based porous materials.
iii. The catalytic active sites are the isolated Me atoms placed in various matrixes, including silicates, LDH or MOFs.
iv. It is widely accepted that the species generated at Me sites upon $H_2O_2$ contact are Me-OOH and Me-$(O_2)$, which are the active species for the oxidative processes.
v. The sulfoxidation reactions are successfully carried out at a low temperature (30–60 °C).
vi. Acetonitrile and methanol appear to be the best solvents for these reactions.
vii. The activity and the stability of catalysts mainly depend on their pore topology and pore size: (a) crystalline Me-silicates are highly active and stable for reactions involving small molecules; (b) mesoporous Me-silicates, which can be used for the sulfoxidation of larger molecules, are less stable in the presence of $H_2O_2$.

Despite the large number of studies and the remarkable body of results gathered in the domain, no industrial sulfoxidation application based on these catalysts was developed until now. The main research challenge is to produce the knowledge necessary to design the "ideal" catalytic system and large-scale oxidation processes.

**Author Contributions:** All authors contributed equally to the work. All authors have read and agreed to the published version of the manuscript.

**Funding:** This research received no external funding.

**Conflicts of Interest:** The authors declare no conflict of interest.

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
