# Peer review of "Mild Oxidation of Organosulfur Compounds with H2O2 over Metal-Containing Microporous and Mesoporous Catalysts"

_catalysts, doi:10.3390/catal11070867_

Round 1

Reviewer 1 Report

This review paper deals with a description of catalytic oxidation of thioeters with H2O2. There are no new experimental data / theoretical considerations presented here, that means the paper presents a description of existing results.

theoretical considerations, to the problem also: models or evaluation of data of other authors.

My suggestion is: reject and resubmit, together with new scientific results.

Author Response

Author response: We regret that the reviewer did not fully understand the aim of this review. In this contribution we examined the representative studies published during the last decades on the sulfoxidation with H2O2 . We believe the originality of our manuscript lies in the description of the main classes of catalysts together (Ti-silicalites, Me-LDH and Me-MOF), as well as the oxidation mechanisms. Our major contributions in the field are underscored in this review.

Reviewer 2 Report

Mild oxidation of organosulfur compounds has drawn great attention in past decades because this reaction can be and has been widely used in fuel refining, organic synthesis, and environmental waste/pollutant treatment. Using H2O2 as the oxidants with heterogeneous catalysts is considered an economical and environment-friendly pathway. In this presented review draft, the authors intended to review relevant scientific aspects concerning the catalysts and the oxidation of thioethers/thiophenes with H2O2. This draft is neatly arranged and presents a reasonable discussion. However, there is still something nonideal for a review paper. Here are the comments:

  1. There are some tying errors and grammar mistakes. For example, line 150 contains a notable red review tracking record. Moreover, some sentences are pretty hard to follow. Please revise those long sentences and check all the typing and grammar.
  2. The authors tried to discuss the mechanism of the catalytic process. They provided an excellent example in chapter 2, where both the experimental results and theoretical estimations are compared and discussed. However, I failed to see more discussion on the theoretical calculation, which has become essential in today’s catalytic mechanism research, in the following chapter. I hope the authors could provide some information about this.
  3. The out-of-date and inappropriate citation. For a review paper in 2021, only 26 of the 146 references (18%) were published in the past five years (2017-2021). If this topic is really important, there must be plenty of relevant research recently. 

In conclusion, I do not think this draft can be published in its current form. So, I cannot recommend that this draft be published without an entire revision.

Author Response

Mild oxidation of organosulfur compounds has drawn great attention in past decades because this reaction can be and has been widely used in fuel refining, organic synthesis, and environmental waste/pollutant treatment. Using H2O2 as the oxidants with heterogeneous catalysts is considered an economical and environment-friendly pathway. In this presented review draft, the authors intended to review relevant scientific aspects concerning the catalysts and the oxidation of thioethers/thiophenes with H2O2. This draft is neatly arranged and presents a reasonable discussion.

Author response: The authors would like to thank the reviewer for the positive endorsement of the work and pertinent comments that have been addressed in the revised version.

However, there is still something nonideal for a review paper. Here are the comments:

  1. There are some tying errors and grammar mistakes. For example, line 150 contains a notable red review tracking record. Moreover, some sentences are pretty hard to follow. Please revise those long sentences and check all the typing and grammar.

Author response: We regret the mistakes. We tried to suppress them in the revised version.

  1. The authors tried to discuss the mechanism of the catalytic process. They provided an excellent example in chapter 2, where both the experimental results and theoretical estimations are compared and discussed. However, I failed to see more discussion on the theoretical calculation, which has become essential in today’s catalytic mechanism research, in the following chapter. I hope the authors could provide some information about this.

Author response: We thank Reviewer 2 for this pertinent suggestion. Accordingly, we have added in sections 4 and    information on the computational data, which complete and confirm the experimental results.

Below the new paragraphs added in the revised version of the manuscript.

Section 4.1.

Tozzola et al. [137] carried out a detailed vibrational and computational study in order to describe the active sites formed over Ti-silicate. The vibrational IR and Raman experiments suggested that under neutral conditions, in the presence of H2O2, an unstable Ti complex containing the O-O moiety (plausibly an OOH group) was formed. This species were transformed in basic conditions into a more stable peroxo form, absorbing at 840 cm-1 (Raman) and 836 cm-1 (IR). The assignment of the spectra was confirmed by ab initio calculations and by parallel experiments performed on the structurally similar Ti-free silicalite.

 Section 4.3.

Experimental and computational results published by Limvorapitux et al. [108] revealed that the Zr6-oxo-hydroxo nodes of UiO-66 (Zr-µ1-OH) became Zr-µ1-OOH after reacting with H2O2. The Zr-μ1-OOH species are active in oxidizing the sulfide as well as its sulfoxide product.

Section 5.

Limvorapitux et al. [108] studied the effect of the solvent on the product distribution obtained on UiO-66 MOF catalyst. They found that the sulfoxidation reaction carried out in CH3OH solvent led to higher selectivities for the sulfoxide product while overoxidation to sulfone predominated in CH3CN and CH2Cl2 solvents. Kinetic studies and computational evaluations supported a model where the sulfoxide product can bind to a site adjacent to the active catalyst species in these latter solvents, resulting in higher degrees of overoxidation through increased local concentration. Such an effect was minimized in CH3OH solvent, which can interact more strongly with the open sites on the nodes than sulfoxide does and thus maintain good sulfoxide selectivity.

  1. The out-of-date and inappropriate citation. For a review paper in 2021, only 26 of the 146 references (18%) were published in the past five years (2017-2021). If this topic is really important, there must be plenty of relevant research recently. 

Author response: Catalytic sulfoxidation with H2O2 is a dynamic topic, which has undergone distinct periods, depending on the nature of the catalysts: Ti-silicalites (1990-2010), LDH (2005-2015) and MOF (2010-2020). In this review we have examined the main aspects concerning the oxidation with H2O2 of sulfides/thiophenes, taking into account all these families of catalysts. For this reason, the list of references covers such an extended period.

Reviewer 3 Report

I have no other comments than some minor spell check is required and, also, please verify that figures and tables are introduced after their first mention in the text.

Author Response

We thank you for your suggestions. In the revised version, we tried to suppress the errors.

Round 2

Reviewer 1 Report

The authors have performed many changes, and the paper can be reconsidered after major revisions, depending on an Editor decision. Please, see listed bellow (questions, Q and remarks, R). The revisions include a) redrawing of figures and b) adding of additional comments concerning optical properties.

Q1 Fig 2 what means mol, % ; y-axe

R1 Fig 3  add y- title; add peak maxima and optical band gap

R2 Fig 4 add optical band gap value, it is an interesting spectrum, and needs comment

R3 Fig 7, y  axis - counts instead of intensity

R4 Fig 8 y axis, c

R5 Fig 9, y axis = F(R)

R6 Fig 10, add peak maxima detected

Author Response

We thank Reviewer for your relevant remarks. Where possible, figures have been modified / corrected

Reviewer 2 Report

After the revision, the current draft is suitable to be published for catalyst.

Author Response

Thank you